# Assessment of Stress and Immune Gene Expression in Australasian Snapper (*Chrysophrys auratus*) Exposed to Chronic Temperature Change

**DOI:** 10.3390/genes16040385

**Published:** 2025-03-28

**Authors:** Kerry Bentley-Hewitt, Christina K. Flammensbeck, Duncan I. Hedderley, Maren Wellenreuther

**Affiliations:** 1The New Zealand Institute for Plant and Food Research Limited, Private Bag 11600, Palmerston North 4442, New Zealand; duncan.hedderley@plantandfood.co.nz; 2The New Zealand Institute for Plant and Food Research Limited, Nelson Research, Centre Box 5114, Port Nelson, Nelson 7043, New Zealand; christina.flammensbeck@plantandfood.co.nz (C.K.F.); maren.wellenreuther@plantandfood.co.nz (M.W.); 3The School of Biological Sciences, The University of Auckland, Private Bag 92019, Auckland 1142, New Zealand

**Keywords:** aquaculture species, snapper, gene expression, genomic, fin-clip, climate change

## Abstract

**Background:** Snapper is a significant commercial, recreational, and cultural teleost species in New Zealand, with aquaculture potential. The impact of long-term (chronic) temperature changes on immune and stress responses have not been studied in snapper, yet they have a critical importance to the health status of the fish. **Methods:** We investigated a set of genes in 30 individual snapper including fin, head kidney, and liver tissue, fish (10 per group) were exposed to either warm (22 °C), cold (14 °C), or ambient temperatures (10.5–18.6 °C) for 3 months. **Results:** Analyses of experimental fish using NanoString technologies to assess stress- and immune-related genes in the three tissue types showed that 22 out of 25 genes changed significantly in the experiment, indicating the significant impacts of chronic temperature changes on stress and immune responses. Furthermore, using a combined dataset based on this study and a previous one testing the impact of acute temperature changes in snapper, we identified five genes in the non-lethal fin-clip samples that can predict internal organ health status. **Conclusions:** Taken together, our experiments demonstrate the potential of the NanoString gene expression assessment tool for the rapid monitoring of stress responses in snapper, which can aid in the selection of stress-resilient wild stocks, monitor species in aquaculture environments, and inform the selection of locations for aquaculture.

## 1. Introduction

Stress responses can be acute or chronic, and temperature changes have a significant impact on these responses in ectotherm animals [1,2,3]. Stress responses are closely linked to immune responses, which are influenced by seasonal changes. For example, temperature increases can enhance immune responses, while temperature decreases have been shown to suppress responsiveness to antigens [4,5,6,7].

The Australasian snapper, *Chrysophrys auratus* (also referred to as tāmure by the indigenous people of New Zealand, family Sparidae, hereafter referred to as snapper), is a species that has been identified as a candidate for aquaculture in New Zealand. The strong potential of snapper for aquaculture comes from the country’s limited diversity of farmed fish species, the need to improve the resilience of the sector, and the challenges posed by a warming climate. Currently, New Zealand’s ocean-based aquaculture is dominated by just three species: Greenshell™ mussel (*Perna canaliculus*), Pacific oyster (*Crassostrea gigas*), and Chinook salmon (*Oncorhynchus tshawytscha*) [8,9,10]. Diversifying the industry is essential for long-term stability and sustainability. A positive factor supporting snapper aquaculture is the well-established product and market already in place. Snapper is one of New Zealand’s largest inshore commercial fisheries and is also highly valued in the country’s significant recreational fishing sector. This existing consumer familiarity and demand provide a strong foundation for expanding into aquaculture. Furthermore, as ocean temperatures rise, selecting species that can thrive in warmer waters is critical. Snapper’s natural resilience to these changing conditions makes it a promising candidate for large-scale production, helping to future-proof New Zealand’s aquaculture industry [11]. 

A selective breeding programme to realise this was started in 2016 [12,13]. Snapper is closely related to red seabream *Pagrus/Chrysophrys major* (hereafter referred to as red seabream) (Temminck & Schlegel 1843), a major aquaculture species in Japan [14]. Breeding of snapper in New Zealand began around 20 years ago at The New Zealand Institute for Plant and Food Research Limited (Plant & Food Research) in Nelson, New Zealand; however, the use of genomic information for selective breeding was initiated only in 2016 to investigate traits of economic interest [15,16]. To support this, our team has developed diverse tools, such as a genome, linkage map, a transcriptome, and genome-wide sequence information of pedigreed snapper, which have been used to improve the breeding of this species [15,17,18,19,20]. 

Global sea temperatures are changing, and this is predicted to have an impact on ectotherms’ stress and immune responses [21,22,23,24]. Previous studies on snapper have shown that high temperatures trigger an immune response [13]. Tolerance to temperature fluctuations is an important trait for the domestication process. The ability to select temperature-tolerant fish is important for breeding. To date there has been no investigation into the impact of chronic long-term (e.g., weeks or months) temperature changes on snapper’s stress and immune responses. Therefore, this study investigated a set of genes identified as potentially being involved in the stress and immune responses of snapper that were exposed to either ambient temperatures (10.5–18.6 °C); a higher temperature from herein called the warm treatment (22 °C); or a lower temperature from herein called the cold treatment (14 °C) for 3 months, from April to July 2020. The study included 10 fish per group (30 in total). Optimal temperatures vary depending on the life stage and the specific phenotypic traits used to define “optimal” conditions. However, for this trial, our focus was on identifying temperatures that promoted optimal survival and growth. These traits are typically maximized at temperatures between 20 °C and 22 °C. The water parameters used in this study were derived from the seasonal minimum and maximum temperatures in the region where the snapper experiments were conducted. Since the Nelson finfish hatchery operates as a flow-through ambient system, water temperatures naturally fluctuate with seasonal changes. Winter temperatures in the area typically reach a minimum of around 10 °C, while summer temperatures peak at approximately 22 °C, with minor variations around these values. The selected experimental temperatures were thus chosen to reflect the natural winter and summer conditions that snapper experience annually.

The genes selected were identified in the snapper transcriptome and were indicated as being involved in stress and immune responses from a literature review focused on red and gilthead seabream (family Sparidae) as established aquacultured and closely related species. The panel of genes was previously used to determine the impacts of acute temperature changes in snapper [25].

Along with determining the impacts of chronic temperature stress on immune- and stress-related genes, we aim to establish non-lethal biomarkers so that we can understand the impacts of temperature changes on snapper. This may lead to the selection of species that can tolerate such changes in temperature extremes, which is critical for maintaining efficient production whilst climate change affects sea temperatures [26,27,28,29]. The skin mucosa contains all innate immune compartments and expresses numerous immune-relevant genes [30], whilst the head kidney and liver have key roles in stress and immune responses [1]. We have focused on all three tissue types, to look for correlations between the tissues and to see if a fin-clip sample could be representative of general immune and stress functions, this being a non-lethal sample for monitoring fish health. For this comparison, we have used data presented in this study and data included in a previous study, which used the same panel of gene targets to increase the power of the data comparisons [25]. Briefly, the previous study explored the set of genes in the fin, head kidney, and liver of juvenile snapper challenged in an acute temperature experiment. Herein, we analysed data from 80 individual snapper challenged in two different temperature experiments (chronic versus acute) by comparing changes in gene expression. Furthermore, we will discuss the comparison of acute and chronic temperature stress by comparing changes in gene expression herein with our previously published data [25].

## 2. Materials and Methods

### 2.1. Experimental Design of Chronic Temperature Stress Trial

All work conducted in this study was approved by the Animal Ethics Committee at the University of Auckland, New Zealand, under ethics approval reference number 002169.

Temperature experiments were conducted at Plant & Food Research’s finfish facility in Nelson, New Zealand. The study population consisted of juvenile snapper that had been generated by the random spawning of 63 adult F_2_ snapper, which were part of a selective breeding programme. This experiment tested **chronic temperature stress** over 3 months. Juvenile snapper (~5 months old at the start of the experiment) were randomly distributed into identical experimental tanks and acclimatised at 18 °C for approx. three weeks. Subsequently, the temperatures were changed to 22 °C for the warm treatment group and to 14 °C for the cold treatment group. The control group was kept at ambient and daily fluctuating temperatures (10.5–18.6 °C) for the whole course of the study (Figure 1). Snapper were fed ad libitum throughout the study. The Plant & Food Research finfish facility operates as a flow-through ambient system, with the water supply remaining at the incoming ambient temperature throughout the study. Temperature measurements were taken twice daily (a.m. and p.m.) with a YSI portable water quality multiparameter instrument. Oxygen was supplemented via a ceramic diffuser to maintain dissolved oxygen levels above 6.5 mg L^−^¹. The pH was regulated between 7.5 and 8.0 through water exchange and aeration. Water quality parameters were monitored twice daily using a YSI Pro1020 Dissolved Oxygen and pH Meter, YSI Incorporated, Yellow Springs, OH, USA. At the end of the experiment after 3 months, fish were anesthetized using AQUI-S^®^ at a concentration of 20 ppm (AQUI-S NZ Ltd., Lower Hutt, New Zealand). This dosage was carefully selected to induce a rapid and consistent anaesthetic effect while allowing for a smooth recovery post-handling. All fish-handling procedures were conducted under anaesthesia to minimize stress and ensure humane treatment. The experiment included eight tanks with 100 fish each. Two tanks were kept at ambient temperature, three tanks were kept at 22 °C, and three at 14 °C. Tissue samples were taken from a subsample of 10 fish per treatment group from the head kidney, liver, and a fin-clip, totalling 30 individuals. This number of animals per subgroup was previously used for our gene expression studies and resulted in significant differences in gene expression [31]. Fish were selected at random for each group, and there were no exclusions of data points (all fish selected were tested). The investigators were not blinded to treatments. Table 1 shows the phenotypic data of each subsample group.

Methods for the **comparative acute temperature experiment** are described in full in our previously published work [25]. Briefly, the acute study explored the same set of genes in the fin, head kidney, and liver tissues of 50 individual fish. Twenty fish were exposed to increasing temperature (up to 31 °C) and 20 fish were exposed to decreasing temperature (down to 7 °C) for up to 37 h. Ten fish were kept at 18 °C (acclimation temperature) as a control group. Thus, in total, data from 80 individual snapper were used for the assessment of whether fin-clip samples could predict internal organ health.

### 2.2. Gene Target Sequences

Twenty-eight genes, including three reference genes, were analysed in fin, head kidney, and liver tissues from an original set of 48 genes included in the panel used for testing. A full description of the design process was previously described in Bentley-Hewitt et al. [25]. 

### 2.3. RNA Extraction

Snap-frozen fin, head kidney, and liver tissues (approximately 15–25 mg) were extracted using the NucleoSpin^®^ RNA, Mini kit, cat. no. 740955.50 (MACHEREY-NAGEL GmbH & Co. KG, Düren, Germany) as previously described in Bentley-Hewitt et al. [25].

### 2.4. Gene Expression Analysis—NanoString nCounter Analysis System

RNA samples (64 ng fin-clip, 21 ng head kidney and 45 ng liver) were analysed using the nCounter Plexset reagents (NanoString Technologies, Inc., Seattle, WA, USA) following the manufacturer’s instructions. Target sequences were designed by NanoString Technologies, Inc. and ordered from Integrated DNA Technologies, Inc., Coralville, IA, USA. Details of target sequences are shown in Appendix A. The amount of RNA added to each NanoString cartridge was limited to a number of gene counts, to avoid overloading of cartridges. The maximum RNA concentration for each tissue was used. Raw gene counts were analysed using the NanoString nCounter Analysis System. Raw counts were normalized using positive controls, and target genes were normalized to the internal reference genes: 40S ribosomal protein S18 (*rps18*), 60S ribosomal protein L8 (*rpl8*), and elongation factor 1-α (*ef1a*), using nSolver™ 4.0 analysis software (NanoString Technologies, Inc.).

### 2.5. Statistical Analyses

Fish phenotypic growth data were analysed using one-way analysis of variance (ANOVA) and Fisher’s least significant differences were calculated post hoc to compare treatment groups. ANOVA of the logs of gene expression counts were used to summarize the data from the fin-clips, head kidney, and liver samples. Fish was fitted as a random effect, and treatment, tissue type, and their interaction as fixed effects. Fisher’s least significant differences were also calculated post hoc between the log-transformed expressions for different genes to compare treatment groups with the control group of each experiment. To adjust for multiple testing, the significance level was set to 0.001 (=0.05/(2 × 25)). Additionally, Pearson correlations between tissue types for each fish were calculated, and included the same panel of genes in the same tissue types from 50 fish that were part of an experiment to test acute temperature changes [25]. The analyses were carried out using Genstat 20th edition (VSN International, Hemel Hempstead, UK) [32].

## 3. Results

### 3.1. Fish Growth and Behaviour in the Chronic Temperature Stress Experiment

Snapper were sampled at three timepoints during the experiment. There were no large changes in fork length or weight between cold and ambient treatment groups. In comparison, fish grew in length (*p* < 0.001) and weighed more (*p* < 0.001) in the warm treatment group than the control fish at both mid-term and final sampling stages. Furthermore, we found that fish in the warm treatment group showed schooling behaviour throughout the experiment and were active during feeding events. In contrast, fish in the ambient and cold groups reduced their food intake and greatly reduced their movements, becoming almost stationary.

### 3.2. Gene Expression in the Chronic Temperature Stress Experiment

Mean gene counts for snapper exposed to either warm (22 °C), cold (14 °C), or ambient temperatures for 3 months are shown in Appendix A, and ranged from 0 to 99,493. There were 22 significant changes in genes when comparing warm- and cold-treated fish to the control group (Figure 2). Gene *socs3* consistently increased in all tissues tested with warm treatment compared with the ambient temperature treatment group, whereas there were no consistent changes in genes across all tissues for the cold treatment compared with the ambient temperature treatment.

For many genes, there were upregulations in liver expression with both warm and cold treatments, compared with the ambient temperature treatment. For example, nine genes (*cat*, *muc18-like*, *cry1-like*, *gsta*, *nrf2*, *prdx1*, *prdx-like*, *sod1*, and *tbfb1-like*) were all upregulated in fish exposed to warm- and cold-water treatments. There were also some consistencies in fin expression when fish were exposed to either warm or cold treatments compared with the ambient temperature treatment. However, *c8a* and *hsp70* were all downregulated in expression, whilst *gapdh* and *igf2* increased in expression.

### 3.3. Correlations of Liver and Head Kidney Gene Expression with Fin Gene Expression

Pearson correlations between tissue gene expression counts were explored in individual fish tissues (Appendix A). The dataset includes the experiment described here in addition to 50 fish from our previous experiment that explored the impact of acute temperature changes over a short period (33–37 h) [25]. There were strong positive correlations between fin tissue and head kidney gene expression for *actb*, *muc18-like*, *cry1-like*, *hsp70*, *prdx5*, and *soc3* in the chronic temperature stress experiment, and a negative correlation with *hsp90*. There were also strong positive correlations between fin and liver for *actb*, *muc18-like*, *hsp70*, *igf1*, *prdx5*, *soc3*, and *gapdh*, and negative correlations for *c8a*, *cry1-like*, *hsp90*, *prdx1-like*, and *sod*. In the acute temperature stress experiment, there were positive correlations between fin and head kidney for *cry1-like*, *hsp70*, *hsp90*, *igf1*, *prdx1*, *prdx5*, and a negative correlation with *cat*. Additionally, positive correlations were shown between fin-clip and liver for *actb*, *cry1-like*, *gsta*, *hsp70*, *hsp90*, *igf1*, and *prdx5*. Therefore, the genes that consistently correlated with fin-clip samples across both experiments were *cry1-like*, *hsp70*, and *prdx5* in the head kidney, and *actb*, *hsp70*, *igf1*, and *prdx5* in the liver (Appendix A).

### 3.4. Comparison of Gene Expression Between Chronic and Acute Exposure to Temperature Differences

Significant differences in gene counts in the warm/heating and cold/cooling treatments across both chronic and acute temperature stress experiments were analysed. There were contrasting responses between chronic thermal stress and acute thermal stress for some genes (summarized in Figure 3). For example, *muc18-like* was upregulated in all tissues in the warm treatment compared with the ambient temperature treatment during the chronic temperature stress trial. However, in the acute temperature stress experiment, the heating treatment downregulated *muc18-like* in the fin and head kidney relative to the control samples, except for the heat-tolerant fish, which maintained their *muc18-like* expression in fin tissues. Similarly, in the acute temperature stress trial, the heating treatment downregulated *cry1-like* in all tissues compared with the control group, whereas the warm treatment increased *cry1-like* in the liver and head kidney from the chronic temperature stress experiment compared with the ambient temperature treatment group. Additionally, fin *hsp70* expression was downregulated in all tissues with the warm and cold treatment compared with the ambient temperature control group in the chronic temperature stress experiment. In contrast, in the acute temperature stress experiment, *hsp70* was upregulated in fin samples with both heating and cooling treatments compared with the control. Furthermore, *prdx1* was downregulated in fin samples with both warm and cold treatments compared with the ambient temperature treatment in the chronic temperature stress trial. However, *prdx1* was upregulated in fin samples with the heating treatment compared with the control during the acute temperature stress experiment. Lastly, *sod1* was upregulated in the liver samples with warm and cold treatments compared with the control during the chronic temperature stress experiment; but in the acute temperature stress experiment, *sod1* was downregulated in the liver and head kidney samples with heating and cooling treatments when compared with control fish.

## 4. Discussion

We utilised a set of stress and immune genes for the analysis of expression in snapper fin, liver, and head kidney tissues to rapidly detect signs of stress following exposure to chronic temperature changes. Additionally, we explored these genes across two experiments to investigate correlations in gene expression between the tissue types, to determine whether a non-lethal fin-clip sample expressed potential health monitoring biomarkers that could be representative of overall fish health (internal and external organ health). Further, we investigated these genes to observe the potential stress and immune gene changes in response to chronic temperature changes and compared these changes to those found following snapper exposed to acute temperature changes described in Bentley–Hewitt et al. [25].

Firstly, we detected 22 significant changes in expression for snapper exposed to either warm (22 °C) or cold (14 °C) compared with ambient temperatures for three months. Because of the number of genes that changed expression levels with the warm and cold treatments compared with the ambient temperature treatment across all tissue types, it was difficult to observe consistent trends in the dataset for some genes. However, this demonstrates the sensitivity of gene expression as a tool for monitoring the effects of chronic (long-term) changes in water temperature. The morphological data (Table 1) documenting changes in weight and fork length highlighted that warmer temperatures are beneficial for snapper growth, in line with previous work on this species. Furthermore, we found that fish in the warm treatment group showed schooling behaviour throughout the experiment and were active during feeding events (fish from all treatment groups were fed ad libitum). In contrast, fish in the ambient and cold groups reduced their food intake and greatly reduced their movements, becoming almost stationary. These behavioural observations suggest a slowing of metabolism to conserve energy. Acceleration of weight gain with warmer temperature treatments has been previously observed in snapper in an experimental setting [18]. It is difficult to speculate how much these differences in growth rate between treatment groups may have affected gene expression. However, it was notable that *igf2* was significantly upregulated in the livers of snapper in the warm treatment group compared to those in the cold and ambient treatment groups. This gene is strongly correlated with the regulation of metabolism and is a growth factor in teleost fish [33].

Secondly, correlations were investigated between tissue gene expression counts in individual fish from this study and from a study examining the effects of acute temperature stress on the same panel of genes and the same tissue samples [25]. Genes that consistently correlated with fin-clip samples in both experiments were *cry1-like*, *hsp70*, and *prdx5* in the head kidney, and *actb*, *hsp70*, *igf1*, and *prdx5* in the liver. This means that these genes may be valuable biomarkers in fin samples to assess health-related responses in key internal organs that play a critical role. The genes that correlated with the non-lethal fin-clip sample in both the head kidney and liver (*hsp70* and *prdx5*) are both involved in protecting against oxidative stress. For example, HSP70 is involved in cytoprotection, cell survival, and immune responses, and increases during inflammation to avoid the apoptosis process [34]. Furthermore, *hsp70* has previously been shown to be upregulated with extreme temperature changes in snapper white muscle, which resulted in protection from oxidative stress and apoptosis [19]. PRDX is involved in the maintenance of oxidative and anti-oxidative processes inside cells, and *pdrx5* was previously shown to be downregulated in the head kidney of gilthead seabream fed polyvinylchloride microparticles [35]. Interestingly, there were contrasting differences in *hsp70* expression between the two experiments, with the warm and cold treatments downregulating *hsp70* relative to the control in the chronic temperature stress experiment, and the heating treatments upregulating *hsp70* relative to the control in the acute temperature stress experiment. This may indicate that *hsp70* is able to respond quickly to acute heating to protect the fish. However, once the temperature changes become chronic, the *hsp70* protection mechanism declines. Despite this, *prdx5* expression was less affected by temperature changes across chronic and acute temperature changes. However, there was a downregulation in *prdx5* expression in the fin following the chronic heat treatment compared with the ambient control, and a downregulation in the liver following the acute heat treatment compared with the control, indicating a consistent reduction of the gene in both experiments as water temperatures increased. This may result in reduced protection against oxidative stress. More studies that explore different experimental conditions and populations of snapper are needed to determine the strength of the correlation under a wider variable spectrum. The fin tissue in fish is in direct contact with the external aquatic environment and can be sampled both rapidly and non-lethally, thus making a fin-clip sample a potentially valuable tissue for population-scale monitoring approaches [36]. Previous research in Japanese ricefish (*Oryzias latipes*) explored expression of genes in response to hypoxia in fin, gill, liver, and brain tissues using microarrays [37]. Their results showed that the fin tissue expression pattern was different from the internal tissues. Similarly, a study on tropical reef clownfish (*Amphiprion ocellaris*) investigated the effects of elevated temperatures on both HSP70 and ubiquitin proteins and found the temperature impacts to be different between muscle and fin tissue, with no detectable protein changes identified in fin tissue [38]. This appears to be a general finding, as both temperature-related changes and tissue-specific changes in HSP70 production or *hsp70* expression levels are common in animals [39,40,41]. In our current study, we detected pronounced changes in *hsp70* gene expression in fin tissue, which may be related to the fact that we analysed gene expression rather than protein production, which may be more sensitive to detecting temperature effects. The method used in this study (NanoString technology) requires only 50 copies of the gene in a given tissue in a calibration assay to be confidently identified in experimental tissue samples. This provides an incredibly sensitive method for detecting changes in gene copy number. The discovery of genes that correlate between fin tissue and internal organs is therefore a novel finding.

Thirdly, significant differences in gene counts in the warm/heat and cold/cool treatments across both long-term chronic and short-term acute temperature stress experiments were analysed for each individual gene. Overall, we detected a high number of genes with contrasting responses between the chronic thermal stress experiment and the acute thermal stress experiment (e.g., *muc18-like* being upregulated by chronic warm conditions). This suggests that different biological mechanisms regulate stress and immune responses, depending on the type of stress applied (e.g., gradual increase in water temperature versus sudden increase in water temperature) [1,42,43]. This is not unexpected, as it is well documented that general acute stress responses generally increase innate responses, and chronic stress responses generally result in immune suppression and increase the risk of infection [1].

The limitations of the study include the fact that the gene annotations for snapper and gilthead seabream genome are still limited, and that genes with -like in their name have been discussed as if they were that gene. We also note that the study used captive fish from one generation, and that stress responses may differ compared with those of wild fish [44,45]. Identifying genetic traits that can be monitored in non-lethal fin-clip sampling and that can indicate the status of internal organs, such as the liver and head kidney, carries significant applied value to inform aquaculture production that aligns with optimal animal welfare conditions. This knowledge can be used in combination with other knowledge to select snapper that have enhanced production traits and improved resilience to temperature fluctuations, which will make snapper a more commercially viable aquaculture species. For example, identifying snapper that maintain high *hsp70* expression during chronic temperature changes may offer a survival advantage as the climate changes. The current study also presents a proof-of-concept approach that could be used, in a modified way, on other species to assess stress resilience, which is urgently needed to inform conservation and management strategies [28,29,46].

In conclusion, we present a set of 25 stress- and immune-related genes in three tissues, to explore their changes when snapper were exposed to chronic warm and cold temperature treatments for a period of three months. Twenty-two of the 25 genes changed significantly between the treatment groups tested, which indicates the significant impacts of chronic water temperature change on stress and immune responses. We also document five genes that can be monitored in non-lethal fin-clip samples and that can predict internal organ health status. Taken together, the gene expression NanoString tool can be used to rapidly monitor stress in snapper non-lethally, and this tool could be used to select stress-resilient wild populations, to monitor species in aquaculture environments, and inform the selection of suitable locations for aquaculture.

## Figures and Tables

**Figure 1 genes-16-00385-f001:**
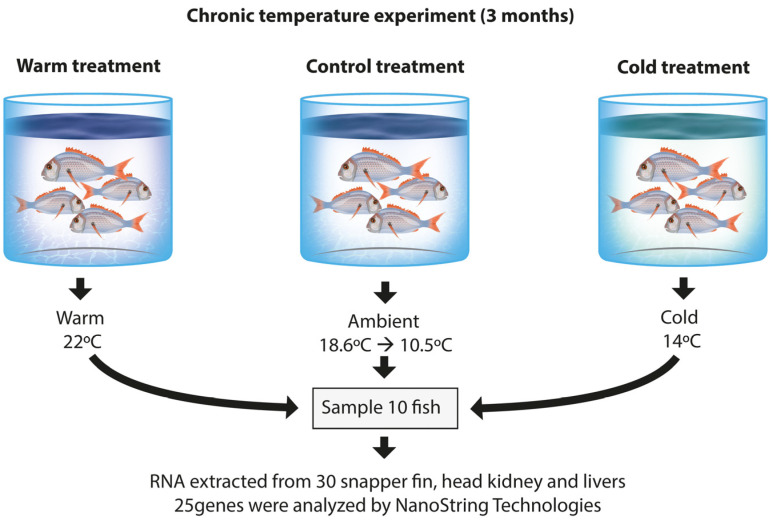
Overview of the chronic temperature stress experimental design.

**Figure 2 genes-16-00385-f002:**
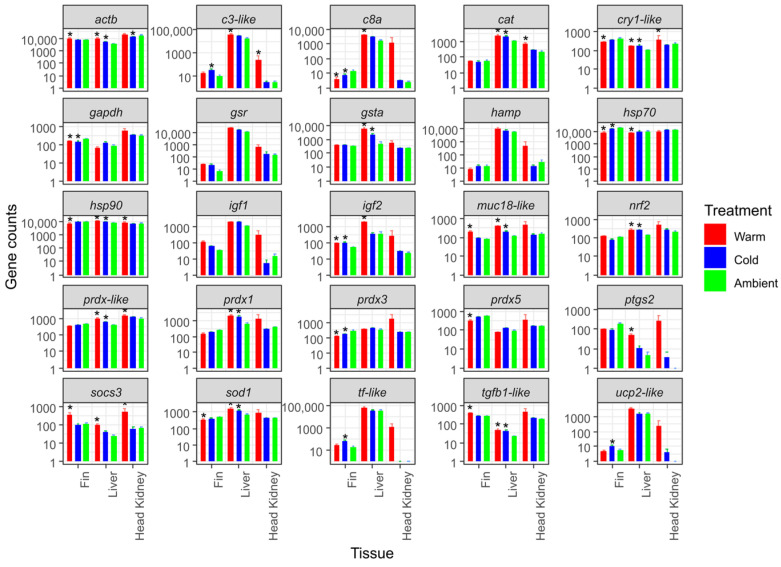
Mean gene counts (*n* = 10) with standard errors (SE) for snapper exposed to either warm (22 °C), cold (14 °C), or ambient temperatures for 3 months. Significant difference between ambient temperature for each tissue is show by *.

**Figure 3 genes-16-00385-f003:**
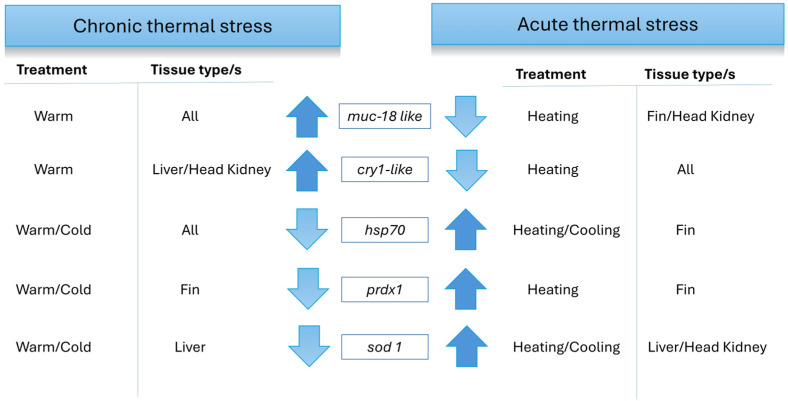
Overview of contrasting snapper gene expression changes when comparing the chronic thermal stress experiment to the acute thermal stress experiment. For each experiment, the treatment and tissue type affected are indicated and the gene changes are indicated by an arrow pointing up for upregulated genes and an arrow pointing down for downregulated genes.

**Table 1 genes-16-00385-t001:** Fork length (mm, mean ± SE) and weight (g, mean ± SE) measurements of 30 juvenile snapper that were sampled for gene expression analysis and exposed to different temperature treatments (warm, cold, and ambient) at three time points throughout the trial (Start of acclimation time (herein referred to initial sampling), Mid-term sampling, and Final sampling).

Treatment Group	Sampling Stage (*n =* 10)	Fork Length [mm]	Weight [g]
Ambient	Initial	82.9 ± 2	11.3 ± 1
	Mid-term	103.4 ± 2	21.7 ± 1
	Final	104.2 ± 2	21.3 ± 1
Warm treatment	Initial	80.1 ± 2	10.4 ± 1
	Mid-term	130.3 ± 3	46.0 ± 4
	Final	158.1 ± 3	88.9 ± 6
Cold treatment	Initial	80.1 ± 5	11.4 ± 2
	Mid-term	99.2 ± 5	21.6 ± 3
	Final	105.2 ± 5	24.8 ± 4

## Data Availability

The data generated and analysed in this study are available upon request from the corresponding author. Access to these data is contingent upon obtaining appropriate consent from the guardians (kaitiaki) of snapper, the indigenous Māori people of New Zealand, to honour their role as stewards of this taonga (treasured species). Potential requestors must provide a detailed explanation of their intended research purpose, including the potential outcomes and benefits of their work. Additionally, requestors are expected to outline any benefit-sharing arrangements before data access can be requested. This ensures that the use and reuse of the data respects the cultural values and guardianship of Māori over natural resources in Aotearoa-New Zealand.

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
