# Peer review of "Assessment of Stress and Immune Gene Expression in Australasian Snapper (Chrysophrys auratus) Exposed to Chronic Temperature Change"

_genes, 2025, doi:10.3390/genes16040385_

Round 1

Reviewer 1 Report (Previous Reviewer 3)

Comments and Suggestions for Authors

Accept in present form

Author Response

Thank you very much for taking the time to review this manuscript. We are pleased you agree it should be accepted for publication.

Reviewer 2 Report (New Reviewer)

Comments and Suggestions for Authors

This paper entitled “Assessment of stress and immune gene expression in Australasian snapper (Chrysophrys auratus) exposed to chronic temperature change” based on heavy lab workload, which is good to point out.

However, firstly, it has serious language problems, and the way to present data is not good enough. Secondly, the scientific target of this study is not clear, this is directly related with how author analyze data and how to discuss the data. Thirdly, the scientific thinking is lacked in this paper, it need to be improved in many aspects.

There are some points offered for improvement in the future:

Line 40:  potential for aquaculture in New Zealand.

Author could give more information to explain why the species Chrysophrys auratus is potential for aquaculture in New Zealand?

Line 52: “do indeed” expressed repeated meaning.

Line 52-56: I suggest to rewrite the sentence.

Line 58: “warm (22°C), cold (14°C)” belong to subjective expression, please try to avoid in scientific papers.

Line 65 “chronic” and “long-term” expressed repeated meaning.

Line 70: The written showing the bold version, please be aware.

Line 94: Same with Line 58, I suggested use “higher and lower” instead.

Line 172: “longer” and “heavier” also belong to subjective expression.

Line 183-189: The result part related with gene expression is scientifically poor written. In gene expression study, we do not present expression level as “Increase, higher and decrease”, we use “upregulated or downregulated” compared with control group.  “Increase and decrease” usually used for when compare the expression dynamic change with time for one specific gene.

Line 197: Author should give what is the statistic method used for correlations study.

Comments on the Quality of English Language

There are too many repeated words and nonscientific words used in this paper. Author could try to use more conjunctions help to improve the logic that could make readers understand more easier. 

Author Response

Response to Reviewer 2 Comments

1. Summary

Comments 1: This paper entitled “Assessment of stress and immune gene expression in Australasian snapper (Chrysophrys auratus) exposed to chronic temperature change” based on heavy lab workload, which is good to point out.

However, firstly, it has serious language problems, and the way to present data is not good enough. Secondly, the scientific target of this study is not clear, this is directly related with how author analyze data and how to discuss the data. Thirdly, the scientific thinking is lacked in this paper, it need to be improved in many aspects.

Response 1: We would like to thank the reviewer for recognizing that this paper is built upon extensive laboratory and experimental work on finfish. We appreciate your acknowledgment of the significant effort invested in generating the data presented in this study.

We also sincerely appreciate your constructive comments on areas that need further improvement. The points you raised regarding language clarity, data presentation, and scientific focus are valuable, and we acknowledge that these aspects require refinement. Specifically, we will work on improving the clarity of our scientific objectives, ensuring that our data analysis and discussion align more effectively with our research aims. Additionally, we will enhance the logical flow and depth of scientific reasoning throughout the manuscript.

Comments 2: Line 40:  potential for aquaculture in New Zealand.

Author could give more information to explain why the species Chrysophrys auratus is potential for aquaculture in New Zealand?

Response 2: We would like to thank the Reviewer for the opportunity to expand. We have added the following text to go into a bit more detail, as requested (Lines 41-54). The text reads as follows: “The strong potential of the snapper for aquaculture comes from the country's limited diversity of farmed fish species, the need to improve the resilience of the sector, and the challenges posed by a warming climate. Currently, New Zealand's ocean-based aquaculture is dominated by just three species: Greenshell™ mussel (Perna canaliculus), Pacific oyster (Crassostrea gigas), and Chinook salmon (Oncorhynchus tshawytscha). Diversifying the industry is essential for long-term stability and sustainability. A positive factor supporting snapper aquaculture is the well-established product and market already in place. Snapper is one of New Zealand’s largest inshore commercial fisheries and is also highly valued in the country’s significant recreational fishing sector. This existing consumer familiarity and demand provide a strong foundation for expanding into aquaculture. Furthermore, as ocean temperatures rise, selecting species that can thrive in warmer waters is critical. Snapper’s natural resilience to these changing conditions makes it a promising candidate for large-scale production, helping to future-proof New Zealand’s aquaculture industry.”

Comments 3: Line 52: “do indeed” expressed repeated meaning.

Response 3: This phrase has been removed from the manuscript (line 66).

Comments 4: Line 52-56: I suggest to rewrite the sentence.

Response 4: These sentences have been reworded (Lines 66-71). The text reads as follows: “Tolerance to temperature fluctuations is an important trait for the domestication process. The ability to select temperature tolerant fish is important for breeding. To date there has been no investigation into the impact of chronic long-term (e.g. weeks or months) temperature changes in snapper on stress and immune responses”

Comments 5: Line 58: “warm (22°C), cold (14°C)” belong to subjective expression, please try to avoid in scientific papers.

Response 5: We have added text to the introduction to clarify the group names (Lines 72-76). The text reads as follows: “This study investigated a set of genes identified as potentially being involved in the stress and immune responses of snapper that were exposed to  ambient temperatures (10.5–18.6°C), a higher temperature from herein called the warm treatment (22°C) or a lower temperature from herein called the cold treatment (14°C) for 3 months, from April to July 2020, which included 10 fish per group (30 in total).”

Comments 6: Line 65 “chronic” and “long-term” expressed repeated meaning.

Response 6: We have removed long-term from the sentence (Line 93).

Comments 7: Line 70: The written showing the bold version, please be aware.

Response 7: We apologise, however we are unsure what the “bold version” is referring to?

Comments 8: Line 94: Same with Line 58, I suggested use “higher and lower” instead.

Response 8:  We have added text to the introduction to clarify the group names (Lines 72-76). The text reads as follows: “This study investigated a set of genes identified as potentially being involved in the stress and immune responses of snapper that were exposed to  ambient temperatures (10.5–18.6°C), a higher temperature from herein called the warm treatment (22°C) or a lower temperature from herein called the cold treatment (14°C) for 3 months, from April to July 2020, which included 10 fish per group (30 in total).” The group names have been retained as warm and cold after the explanation.

Comments 9: Line 172: “longer” and “heavier” also belong to subjective expression.

Response 8:  We have removed the terms longer and heavier from the manuscript (Lines 216-219). The text reads as follows: “In comparison, fish grew in length (P<0.001) and weighed more (P<0.001) in the warm treatment group compared to the control at both mid-term and final sampling stages.”

Comments 10: Line 183-189: The result part related with gene expression is scientifically poor written. In gene expression study, we do not present expression level as “Increase, higher and decrease”, we use “upregulated or downregulated” compared with control group.  “Increase and decrease” usually used for when compare the expression dynamic change with time for one specific gene.

Response 10:. We have edited the manuscript throughout replacing increase and decrease terms with upregulate and downregulate terms when refereeing to gene expression. The includes the section referred to in the comment (now lines 229-236), additionally lines 261-292 and the discussion section.

Comments 11: Line 197: Author should give what is the statistic method used for correlations study.

Response 11: We have added “Pearson” to line 244 to describe the statistical method used.

Comments 12: There are too many repeated words and nonscientific words used in this paper. Author could try to use more conjunctions help to improve the logic that could make readers understand more easier. 

Response 12: We believe we have addressed this issue throughout the manuscript. Details are provided in the responses shown above.

Reviewer 3 Report (New Reviewer)

Comments and Suggestions for Authors

The review report for genes-3507978 below.

Title: Assessment of stress and immune gene expression in Australasian snapper (Chrysophrys auratus) exposed to chronic temperature change.

Along with determining the impacts of chronic long-term temperature stress on immune- and stress-related genes, aim to establish non-lethal biomarkers so that can understand the impact of temperature changes in snapper.

The discovery of genes that correlate between fin tissue and internal organs is it a novel finding.

Identifying genetic traits that can be monitored in non-lethal fin clip sampling that can indicate the status of internal organs, such as the liver and head kidney, carries significant applied value to inform aquaculture production that aligns with optimal animal welfare conditions.

The references cited in this manuscript are appropriate

The tables and figures are clear.

More studies are needed that explore different experimental conditions and populations of snapper to determine the strength of the correlation under a wider variable spectrum.

The lacks informations:

The introduction 

In of this manuscript, there is no information about the optimal water temperature for the tested juvenile fish specie (Australasian snapper) - please add this information.

The methodology

  1. How was the warm (22°C) and cold (14°C) water temperature determined to be exposed to fish in a reviewed manuscript?
  2. What water parameters and with what frequency (once a day, several times a day) were measured during the experience described in this manuscript? Please provide values
  3. Were the fish in the experiment anesthetized or sedated?
  4. What anesthetic was used in the experiment? In what doses?

Please explain it.

Author Response

Response to Reviewer 3 Comments

1. Summary

Comments 1: The review report for genes-3507978 below.

Title: Assessment of stress and immune gene expression in Australasian snapper (Chrysophrys auratus) exposed to chronic temperature change.

Along with determining the impacts of chronic long-term temperature stress on immune- and stress-related genes, aim to establish non-lethal biomarkers so that can understand the impact of temperature changes in snapper.

The discovery of genes that correlate between fin tissue and internal organs is it a novel finding.

Identifying genetic traits that can be monitored in non-lethal fin clip sampling that can indicate the status of internal organs, such as the liver and head kidney, carries significant applied value to inform aquaculture production that aligns with optimal animal welfare conditions.

 The references cited in this manuscript are appropriate

The tables and figures are clear.

More studies are needed that explore different experimental conditions and populations of snapper to determine the strength of the correlation under a wider variable spectrum.

Response 1: Thank you for your comments. We have revised the manuscript and addressed your comments. Detailed are described below.

Comments 2: The lacks informations: The introduction 

In of this manuscript, there is no information about the optimal water temperature for the tested juvenile fish specie (Australasian snapper) - please add this information.

Response 2: We have added more information into the introduction regarding the optimal temperatures for the fish (Lines 76-80). The text reads as follows: “Optimal temperatures vary depending on the life stage and the specific phenotypic traits used to define "optimal" conditions. However, for this trial, our focus was on identifying temperatures that promote optimal survival and growth. These traits are typically maximized at temperatures between 20–22°C.”

Comments 3: The methodology…How was the warm (22°C) and cold (14°C) water temperature determined to be exposed to fish in a reviewed manuscript?

Response 3: We have included more information regarding the choice of water temperatures selected in this study in the introduction (lines 76-87). The text reads as follows: “Optimal temperatures vary depending on the life stage and the specific phenotypic traits used to define "optimal" conditions. However, for this trial, our focus was on identifying temperatures that promote optimal survival and growth. These traits are typically maximized at temperatures between 20–22°C. The water parameters used in this study were derived from the seasonal minimum and maximum temperatures in the region where the snapper experiments were conducted. Since the finfish hatchery operates as a flow-through ambient system, water temperatures naturally fluctuate with seasonal changes. Winter temperatures in the area typically reach a minimum of around 10°C, while summer temperatures peak at approximately 22°C, with minor variations around these values. The selected experimental temperatures were chosen to reflect the natural winter and summer conditions that snapper experience annually.”

Comments 4: What water parameters and with what frequency (once a day, several times a day) were measured during the experience described in this manuscript? Please provide values

Response 4: We have included more details into the methods to describe how the fin fish facility operates (Lines 127-128). Te text reads as follows: “The PFR finfish facility operates as a flow-through ambient system, with the water supply remaining at the incoming ambient temperature throughout the study. “ In addition, we have included more information regarding the monitoring of water (lines 130-133). The text reads as follows:

“Oxygen was supplemented via a ceramic diffuser to maintain dissolved oxygen levels above 6.5 mg L⁻¹. pH was regulated between 7.5 and 8.0 through water exchange and aeration. Water quality parameters were monitored twice daily using a YSI Pro1020 Dissolved Oxygen and pH Meter. “

Comments 5: Were the fish in the experiment anesthetized or sedated?

Response 5: The fish were anesthetized and details have been included in the manuscript (line 134-138). The text reads as follows: “Fish were anesthetized using AQUI-S at a concentration of 20 ppm (AQUI-S NZ Ltd, Lower Hutt, New Zealand). This dosage was carefully selected to induce a rapid and consistent anesthetic effect while allowing for a smooth recovery post-handling. All fish handling procedures were conducted under full anesthesia to minimize stress and ensure humane treatment.”

Comments 6: What anesthetic was used in the experiment? In what doses?

Please explain it.

Response 6:  The information regarding the anesthetic used and the dose is now included in the manuscript (lines 134-138). The text reads as follows: “Fish were anesthetized using AQUI-S at a concentration of 20 ppm (AQUI-S NZ Ltd, Lower Hutt, New Zealand). This dosage was carefully selected to induce a rapid and consistent anesthetic effect while allowing for a smooth recovery post-handling. All fish handling procedures were conducted under full anesthesia to minimize stress and ensure humane treatment.”

Round 2

Reviewer 2 Report (New Reviewer)

Comments and Suggestions for Authors

Author has already changed the little issues mentioned in the last review report.

However, the quality of this paper has not improved significantly, especially the gene counts or gene expression study, which is giving the expression that principle of the gene expression/gene count and logs of gene expression hasn't been presented well in this study. Overall, this study hasn't shown in a scientific and systematic way from a reviewer's point of view. 

Comments on the Quality of English Language

Author has addressed the language problems that mentioned in the last report, but the overall language level is still need to be improve a lot for the scientific publication. 

Author Response

Response to Reviewer 2 Comments

Round 2

Summary

Comments 1: Author has already changed the little issues mentioned in the last review report. However, the quality of this paper has not improved significantly, especially the gene counts or gene expression study, which is giving the expression that principle of the gene expression/gene count and logs of gene expression hasn't been presented well in this study. Overall, this study hasn't shown in a scientific and systematic way from a reviewer's point of view. 

Response 1: All scientific papers must go through an independent and rigorous review process at The New Zealand Institute for Plant and Food Research Limited, before permission is granted to submit them. First, two scientists not connected with the paper must review it. Then the Science Group Leader (not directly connected with the paper) must review it. Finally, a professional Scientific Editor must review it. This process is iterative, with feedback used to improve the manuscript until it is deemed to be of sufficient quality to be submitted for publication.

Changes have been made to this manuscript in response to the reviews by Genes, I have also obtained a review of the latest iteration of the manuscript by another scientific editor at The New Zealand Institute for Plant and Food Research Limited, to catch any problems that may have arisen since the original submission. Her comments are given below:

Dr Anne Gunson, Senior Science Editor and Science Publication Office Manager at the New Zealand Institute for Plant and Food Research Limited on 14th March, 2025.

“I have today read through your submitted manuscript. I have made some edits to the revised text, and I am now confident that it is well written, and reaches the standard needed for good scientific writing. Your rebuttal addresses the reviewers’ comments thoroughly, and I see no reason why the journal editor should not accept this paper for publication.”

 In addition, with have added extra figures to the manuscript to improve the clarity to the reader. This includes (1) a schematic overview of the acute temperature experiment, (2) a figure to show the gene counts from the chronic temperature experiment and (3) a schematic overview of contrasts in gene expression between the acute and chronic thermal stress studies.

Comments 2: Comments on the Quality of English Language

Author has addressed the language problems that mentioned in the last report, but the overall language level is still need to be improve a lot for the scientific publication

Response 2: As mentioned above, the revised version has been reviewed again from our Senior Science editor.

This manuscript is a resubmission of an earlier submission. The following is a list of the peer review reports and author responses from that submission.

Round 1

Reviewer 1 Report

Comments and Suggestions for Authors

The work by Bentley-Hewitt is an interesting study with notable potential practical applications for the commercial culture of Australasian snapper. The work in its current state is a bit of a mess in regards to the Materials and methods. There seems to be a great amount of the information for this paper that is included in the famous “paper 1” referenced repeatedly herein. If there is such an abundance of relevance then more background of some detail should perhaps be included in the Introduction section.

MATERIALS and METHODS

As currently written, it appears that the trial is 30 days of chronic exposure to 3 temperature regimes (cold, warm, and ambient). There is a need for extensive revision of the Materials and Methods to make more clear the difference between the “short term” and “long term” trials; the cooling and heating trials; and the number of genes analyzed (figure 1 it is 25 genes, then line 100 it is 48 genes, then again on lines 264 and 266 it is back to 25 genes!). In Figure 2 there is a description of “heat tolerant”, “heat sensitive”, “cold tolerant”, and “cold sensitive”. This is a reference to a short term trial that appears nowhere in the Materials and Methods! If this is all described in the reference [16] by Bentley-Hewitt it is not sufficient to repeatedly cite this reference to avoid including the necessary text herein if it is needed for following the flow of this work. This work should be able to “stand alone” so to speak.

The gene expression description could benefit from some additional text to state precisely what is the technology used for the gene expression analysis (microarray, qPCR, etc…); simply stating a trademarked name of a company’s service offering, is a sloppy way to write M&M, to say the least.

On line 106 there are different amounts of starting RNA described for each tissue type. What purpose does this serve? Why is the amount of RNA not standardized? There may well be a rationale for this but it should be explained.

RESULTS

The Results section suffers from the same lack of clarity as to its relation to the methods described. Table 1 has only the 3 temperature regimes displayed. Table 2 then appears the “short term” and “long term” experiments that are not described in the M&M.

DISCUSSION

The discussion focuses on specific genes, but if there are cold tolerant strains and heat tolerant strains as suggested by the figure 2 actually exist then this would be of interest for a more expansive part of the Discussion. The authors write (line 239): “…our study analysed gene expression rather than the protein production and this may be more sensitive to detect temperature effects.” This is an interesting observation that is worthy of a deeper exploration in the bibliography to expand on this idea to find what others might have discovered in this area. Also on line 250-252: “This indicates different biological mechanisms are involved in the regulation of stress and immune responses that are dependent on the type of stress applied...” ; this is another topic that could be expanded by a deeper exploration of the bibliography to give the Discussion more interest and relevance.

REFERENCES

Line 60, 110, 126, 168, 213, and 221: “published elsewhere” is not a reference. If it is published, then provide said reference in the text and bibliography. Moreover, if it is actually relevant for understanding the current work, include more text.

Line 222: “Ref paper 1 is not a reference.

LINE NOTES

Line 179: insert space between “graph” and “in”.

Line 186: Remove the extra “chronic”.

Line 186: Insert space between “term” and “thermal”.

Author Response

Response to Reviewer 1 Comments

1. Summary

Comments 1: The work by Bentley-Hewitt is an interesting study with notable potential practical applications for the commercial culture of Australasian snapper. The work in its current state is a bit of a mess in regards to the Materials and methods. There seems to be a great amount of the information for this paper that is included in the famous “paper 1” referenced repeatedly herein. If there is such an abundance of relevance then more background of some detail should perhaps be included in the Introduction section.

Response 1: Thank you for your comment. Paper 1 has since been published and can be referenced accordingly. We have included more detail into the introduction section to describe the brief outline of experimental details for paper 1 in lines 79-87. “. For this comparison we have used data presented in this study and data included in a previous study, which used the same panel of gene targets [24]. Briefly, this study explored the set of candidate genes in the fin, head kidney and liver tissues of 50 individuals by exposing 20 fish to increasing temperature (up to 31 °C) and 20 fish to decreasing temperature (down to 7 °C) for up to 37 h. Of these, we analyzed 10 temperature-sensitive and 10 temperature-tolerant fish, along with 10 fish kept at 18 °C (acclimation temperature) as a control group. Thus in total data from 80 individual snapper were compared. This study represented acute temperature stress.” In addition, we added in details for the number of fish included in the present study in lines 62-63. “which included 10 fish per group (30 total).”

Comments 2: MATERIALS and METHODS

As currently written, it appears that the trial is 30 days of chronic exposure to 3 temperature regimes (cold, warm, and ambient). There is a need for extensive revision of the Materials and Methods to make more clear the difference between the “short term” and “long term” trials; the cooling and heating trials; and the number of genes analyzed (figure 1 it is 25 genes, then line 100 it is 48 genes, then again on lines 264 and 266 it is back to 25 genes!). In Figure 2 there is a description of “heat tolerant”, “heat sensitive”, “cold tolerant”, and “cold sensitive”. This is a reference to a short term trial that appears nowhere in the Materials and Methods! If this is all described in the reference [16] by Bentley-Hewitt it is not sufficient to repeatedly cite this reference to avoid including the necessary text herein if it is needed for following the flow of this work. This work should be able to “stand alone” so to speak.

The gene expression description could benefit from some additional text to state precisely what is the technology used for the gene expression analysis (microarray, qPCR, etc…); simply stating a trademarked name of a company’s service offering, is a sloppy way to write M&M, to say the least.

On line 106 there are different amounts of starting RNA described for each tissue type. What purpose does this serve? Why is the amount of RNA not standardized? There may well be a rationale for this but it should be explained.

Response 2: Thank you for your comments. For clarity we have now standardized the names of long-term and short-term study to chronic and acute throughout the manuscript. We have clarified the discrepancy in the number of target genes finally analysed in section 2.2 line 133-142 by giving more details on the process of selecting the gene candidates and why some had to be excluded from the final analysis. This now reads “Forty eight candidate genes, including reference genes were selected from a literature review of studies looking at immune- and stress-related genes, mainly in teleost fishes from the family Sparidae such as red and gilthead seabream, and whether they could potentially be identified in the snapper transcriptome. Details of the literature searches and the design process are described in Bentley-Hewitt et al. [24]. However, only 28 genes passed calibration due to the expression of genes counts varying considerably, and to avoid overloading the NanoString cartridges, the concentration of RNA added was low. This resulted in many genes on the panel having low counts, which could not be calibrated by the assay’s nSolver™ 2.0 software, as each gene requires 50 counts in the eight lanes containing the calibration sample. These genes had to be removed from the analysis, leaving 25 target genes and three reference genes. One of the initial reference genes gapdh was removed because of its variability in the samples, and it was treated as a target gene. The final target genes were normalized to three reference genes that were stable and passed calibration as described in section 2.4.” In addition, we have included a reference to supplementary table 1 in lines 158-159. This table includes full details of the 48 gene targets and indicates which are reference genes, target genes that passed calibration and target genes that did not pass calibration.

We have included more details for the acute temperature experiment in section 2.1 that explains the temperature tolerant and temperature sensitive labels in figure 2 in lines 109-115. This now reads “ Fish rearing for the study population and full details of the acute temperature experiment have been previously described in Bentley-Hewitt et al. [24]. Briefly this study explored the set of candidate genes in the fin, head kidney and liver tissues of 50 individuals by exposing 20 fish to increasing temperature (up to 31 °C) and 20 fish to decreasing temperature (down to 7 °C) for up to 37 h. Of these 10 temperature-sensitive and 10 temperature-tolerant fish, along with 10 fish kept at 18 °C (acclimation temperature) as a control group were analysed. Thus in total data from 80 individual snapper were compared.”

The technology used for gene expression analysis is called NanoString technology as described in this manuscript. We have included more detail as to why there is a different amount of RNA loaded onto the cartridges used in this technology included in section 2.4 lines 159-164. This reads The amount of RNA added to each NanoString cartridge is limited to a number of gene counts to avoid overloading of cartridges. The maximum RNA concentration for each tissue was used.”

Additionally we have included a new table 1 showing the phenotypic data on the different sample groups referenced in lines 107-108.

Comments 3: RESULTS

The Results section suffers from the same lack of clarity as to its relation to the methods described. Table 1 has only the 3 temperature regimes displayed. Table 2 then appears the “short term” and “long term” experiments that are not described in the M&M.

Response 3: Thank you for your comments. For clarity we have now standardized the names of long-term and short-term study to chronic- and acute- temperature experiments throughout the manuscript. This includes table 2 and figure 2. Additionally, more details of the acute (formally named short-term) experimental design have been included in the material and methods section.

Comments 4: DISCUSSION

The discussion focuses on specific genes, but if there are cold tolerant strains and heat tolerant strains as suggested by the figure 2 actually exist then this would be of interest for a more expansive part of the Discussion. The authors write (line 239): “…our study analysed gene expression rather than the protein production and this may be more sensitive to detect temperature effects.” This is an interesting observation that is worthy of a deeper exploration in the bibliography to expand on this idea to find what others might have discovered in this area. Also on line 250-252: “This indicates different biological mechanisms are involved in the regulation of stress and immune responses that are dependent on the type of stress applied...” ; this is another topic that could be expanded by a deeper exploration of the bibliography to give the Discussion more interest and relevance.

Response 4: The details of genes that may confer a more tolerant phenotype towards acute temperature stress is included in our previously published study. The key aim for this study to explore a non-lethal biomarker assessment for fish health (the fin clip sample) and the effects of chronic temperature changes on potential gene biomarkers. In this manuscript we do include and briefly discuss the advantages of selecting genetic traits (lines 352-359) and have added in an example. This section now readsThis knowledge can be used in combination with other knowledge to selectively breed snapper for enhanced production traits while improving resilience to temperature fluctuations and ultimately to make this a more commercially viable aquaculture species. For example, identifying snapper that maintain high hsp70 expression during chronic temperature changes may offer a survival advantage as the climate changes. The study here also presents a proof-of-concept approach that could be used, in a modified way, in other species to assess stress resilience, something that is urgently needed to inform conservation and management strategies [28,29,45].” We have expanded on the potential reasoning as to why we saw changes in fin gene expression when another study did not show difference in the protein production in fin. We have included (lines 324-328) “The methodology used in this study (NanoString technology) requires only 50 copies of the gene in a specific tissue in a calibration test to be identified confidently in experimental tissue samples. This offers an incredible sensitive method for detection of changes in gene copy number.” Additionally, we have expanded the discussion to explain that the contrasting results since in our study between chronic and acute stress is expected (Lines 340-343). “This is not unexpected as it is well documented that in general stress responses involve the neural, endocrine and immune systems and can be acute and generally increase innate responses or chronic which can result in immune-suppressive responses and increased the risk of infection [1].”

Comments 4: REFERENCES

Line 60, 110, 126, 168, 213, and 221: “published elsewhere” is not a reference. If it is published, then provide said reference in the text and bibliography. Moreover, if it is actually relevant for understanding the current work, include more text.

Line 222: “Ref paper 1 is not a reference.

LINE NOTES

Line 179: insert space between “graph” and “in”.

Line 186: Remove the extra “chronic”.

Line 186: Insert space between “term” and “thermal”.

Response 4: Thank you for your comments we have removed published elsewhere throughout the manuscript and fully referenced the article in question as this has now been published. Additionally we have tided up the document to remove the errors and provided a full reference when we had formally stated Ref paper 1.

Reviewer 2 Report

Comments and Suggestions for Authors

In the present manuscript the authors analyzed the expression a panel of 25 immune related genes in different tissue of Australasian snapper after a long- term temperature challenge exposure.

The idea is interesting and a possible correlation between the expression of immune related genes, which can constitute a non-invasive marker to assess the stress condition of the animals is valuable. However, I found numerous inaccuracies and shortcomings, both methodological, conceptual and in the development of the paper. From a general perspective this manuscript seems like a small appendix of the previous, and already published, paper of the same author (Bentley-Hewitt et al., 2024). Therefore, there is important information that is completely lacking, such as the ethical evaluation of the experimental design, the complete list of the genes analyzed, with their relative sequences and access numbers, a proper statistical deep evaluation of the results (such as the ANOVA results, as the test was mentioned in the M&M, but the results are absent). I can understand that this info is reported in the previous paper, but the authors should briefly mention them also in this work to allow the reader to fully understand the results with all possible tools.

Another problem I can see from this paper is that apart from Figure 1, which is the graphical representation of the experimental design, two out of three graphical materials of the present manuscript are based on the results obtained in another paper, and so not generated within this project. I can understand that the authors with this work aimed to confirm an assumption previously obtained, but in this way, the present manuscript does not have its own entity and the results generated by it do not support the conclusions expressed, if not marginally and partially.

Finally, I suggest to the authors to further contextualize the correlations and results obtained with published bibliography. Additionally, I also suggest to discuss the present results from a more physiological and functional point of view with respect to the animal (such as lines 264-266), things that I have not seen in the discussion, which appears to be lacking in depth and content.

Therefore, as a conclusion, I suggest the author strongly reconsider the organization of the manuscript, including further experimental data, such as the growth performance of the animals, their behavior, the environmental conditions associated with the changes in the water temperature, and the histological assessment which can corroborate the potential stress condition of the animals. I just gave some examples to make this work more solid and with its own strength.

Author Response

Response to Reviewer 2 Comments

1. Summary

Comments 1: In the present manuscript the authors analyzed the expression a panel of 25 immune related genes in different tissue of Australasian snapper after a long- term temperature challenge exposure.

The idea is interesting and a possible correlation between the expression of immune related genes, which can constitute a non-invasive marker to assess the stress condition of the animals is valuable.

Response 1: Thank you for your positive comment on our study. We appreciate you highlighting that our work offers a valuable insight into the potential of a non-invasive biomarker to monitor fish health.

Comments 2: However, I found numerous inaccuracies and shortcomings, both methodological, conceptual and in the development of the paper. From a general perspective this manuscript seems like a small appendix of the previous, and already published, paper of the same author (Bentley-Hewitt et al., 2024). Therefore, there is important information that is completely lacking, such as the ethical evaluation of the experimental design, the complete list of the genes analyzed, with their relative sequences and access numbers, a proper statistical deep evaluation of the results (such as the ANOVA results, as the test was mentioned in the M&M, but the results are absent). I can understand that this info is reported in the previous paper, but the authors should briefly mention them also in this work to allow the reader to fully understand the results with all possible tools.

Response 2: Thank you for your comments. We have now included more information on the additional study data used for some of the analysis that was previously published in section 2.1 lines 109-115. This now reads “ Fish rearing for the study population and full details of the acute temperature experiment have been previously described in Bentley-Hewitt et al. [24]. Briefly this study explored the set of candidate genes in the fin, head kidney and liver tissues of 50 individuals by exposing 20 fish to increasing temperature (up to 31 °C) and 20 fish to decreasing temperature (down to 7 °C) for up to 37 h. Of these 10 temperature-sensitive and 10 temperature-tolerant fish, along with 10 fish kept at 18 °C (acclimation temperature) as a control group were analysed. Thus in total data from 80 individual snapper were compared.” Additionally, we have included reference to the ethical approval for this study lines 94-96 “All work conducted in this study was approved by the Animal Ethics Committee at the University of Auckland, New Zealand, under ethics approval reference number 002169.” We have included a full list of genes analysed, with their relative sequences and access numbers in supplementary table 1 referenced on lines 158-159. The results of the statistical tests are presented in the tables and graph. We hope that the changes made have given more clarity to the reader.

Comments 3: Another problem I can see from this paper is that apart from Figure 1, which is the graphical representation of the experimental design, two out of three graphical materials of the present manuscript are based on the results obtained in another paper, and so not generated within this project. I can understand that the authors with this work aimed to confirm an assumption previously obtained, but in this way, the present manuscript does not have its own entity and the results generated by it do not support the conclusions expressed, if not marginally and partially.

Response 3: Thank you for your comment. To clarify, table 2 (formally table 1) presents the results from this study only. Table 3 (formally table 2) and figure 2 presents results which include both data from the current study and the additional study previously published. The reason is to improve the statistical strength of the study in looking for correlations in tissue gene expression taking the n number from 30 fish (current study) to 80 individual fish.  

Comments 4: Finally, I suggest to the authors to further contextualize the correlations and results obtained with published bibliography. Additionally, I also suggest to discuss the present results from a more physiological and functional point of view with respect to the animal (such as lines 264-266), things that I have not seen in the discussion, which appears to be lacking in depth and content.

Response 4: Thank you for your comment. We have expanded the discussion significantly and discuss the relevance of the results in a more physiological functional view point. Specifically in lines “288-308”, which now reads The genes that correlated across both head kidney and liver with the non-lethal fin clip sample (hsp70 and prdx5) are both involved in oxidative stress protection. For example, HSP70 is involved in cytoprotection, cell survival and immune responses. It is inducible throughout inflammation and represents an effort to avoid apoptosis [32]. hsp 70 increased at temperature extremes in snapper white muscle and offered protection from oxidative stress and apoptosis [18]. Whilst, PRDX is involved in the maintenance of oxidative and anti-oxidative processes inside cells and pdrx5 decreased in the head kidney of gilthead seabream fed polyvinylchloride microparticles [33]. Interestingly, there were contrasting differences between hsp70 expression between both experiments whereas warm and cold treatments decreased hsp70 compared to the control in the chronic-temperature experiment, whilst the heating treatments increased hsp70 compared to the control in the acute-temperature experiment. This may demonstrate that hsp70 is able to respond rapidly with acute heating to help protect the fish. However, once the temperature changes become chronic hsp70 protection mechanism decline. Despite this prdx5 expression was less effected by temperature changes across chronic- and acute- temperature changes. However, there was a decrease in prdx5 expression in the fin following the chronic warm treatment compared to the ambient control and a decrease in the liver following acute heating treatment compared to the experimental control, indicating a consistent reduction in the gene across both experiments when water temperatures increase that may result in decreased protection from oxidative stress.”

Comments 5: Therefore, as a conclusion, I suggest the author strongly reconsider the organization of the manuscript, including further experimental data, such as the growth performance of the animals, their behavior, the environmental conditions associated with the changes in the water temperature, and the histological assessment which can corroborate the potential stress condition of the animals. I just gave some examples to make this work more solid and with its own strength.

Response 5: Thank you for your comment. We have made considerable edits to the manuscript to improve the clarity and information given to the reader as highlighted throughout the document in track changes. Based on the reviewer’s feedback, we have now also included a new table 1 showing the phenotypic data on the different snapper sample groups referenced in lines 107-108. This was done to give the reader more detail on the growth of fish during the study.  

Reviewer 3 Report

Comments and Suggestions for Authors

The study assesses the impact of chronic temperature variations on the stress and immune responses in Australasian snapper (Chrysophrys auratus) by analyzing gene expression in fin, head kidney, and liver tissues. Over three months, juvenile snapper were subjected to warm (22°C), cold (14°C), and ambient (10.5–18.6°C) conditions. Using NanoString technology, the researchers monitored a set of 25 candidate genes, finding that 22 showed significant expression changes under temperature stress. The study highlights five genes as reliable biomarkers for non-lethal health assessments via fin clips, potentially predicting internal organ status. This approach may aid aquaculture by enabling early stress detection, supporting resilient population management, and informing optimal habitat selection amid climate change.

The use of NanoString technology for gene expression analysis ensures precise and detailed quantification of stress and immune markers, providing reliable and reproducible results. Additionally, the study’s comprehensive analysis across multiple tissues (fin, head kidney, and liver) enables a holistic understanding of the physiological impacts of thermal stress, further validating the use of fin samples as proxies for internal health status. Moreover, the study holds high relevance for the aquaculture industry, as it identifies genes associated with resilience to thermal stress. This has the potential to inform selective breeding strategies, supporting the development of more resilient fish populations in the face of climate change. The limitations of the study are well-discussed. While targeted, the panel of 25 genes selected for the analysis may not fully capture the complexity of immune and stress responses, potentially omitting additional genes that could be critical for understanding the broader physiological impacts of temperature changes on snapper. Additionally, the study focuses exclusively on thermal stress, which is indeed a significant factor in aquaculture, but does not consider other common stressors such as water quality, population density, or nutrition. In practical aquaculture settings, these factors often interact, influencing fish health in complex ways that may limit the direct applicability of the findings under multifactorial conditions. Moreover, the study observes fish responses over a three-month period, leaving open questions about the persistence of these effects. A longer-term follow-up could reveal whether chronic thermal stress has lasting impacts on gene expression or if the fish can fully recover once conditions return to normal. Extending the observation period might also provide insights into the potential for adaptation or resilience in response to prolonged stress. However, this study is methodologically robust and well-argued. While it focuses on a specific gene panel and thermal stress alone over a limited period, the Authors address these limitations effectively, providing a balanced and thoughtful discussion. Overall, this work lays a solid foundation for future studies to broaden genetic markers and environmental factors, making it a meaningful contribution to aquaculture research.

Author Response

Response to Reviewer 3 Comments

1. Summary

Thank you very much for taking the time to review this manuscript.

Comments 1: The study assesses the impact of chronic temperature variations on the stress and immune responses in Australasian snapper (Chrysophrys auratus) by analyzing gene expression in fin, head kidney, and liver tissues. Over three months, juvenile snapper were subjected to warm (22°C), cold (14°C), and ambient (10.5–18.6°C) conditions. Using NanoString technology, the researchers monitored a set of 25 candidate genes, finding that 22 showed significant expression changes under temperature stress. The study highlights five genes as reliable biomarkers for non-lethal health assessments via fin clips, potentially predicting internal organ status. This approach may aid aquaculture by enabling early stress detection, supporting resilient population management, and informing optimal habitat selection amid climate change.

The use of NanoString technology for gene expression analysis ensures precise and detailed quantification of stress and immune markers, providing reliable and reproducible results. Additionally, the study’s comprehensive analysis across multiple tissues (fin, head kidney, and liver) enables a holistic understanding of the physiological impacts of thermal stress, further validating the use of fin samples as proxies for internal health status. Moreover, the study holds high relevance for the aquaculture industry, as it identifies genes associated with resilience to thermal stress. This has the potential to inform selective breeding strategies, supporting the development of more resilient fish populations in the face of climate change. The limitations of the study are well-discussed. While targeted, the panel of 25 genes selected for the analysis may not fully capture the complexity of immune and stress responses, potentially omitting additional genes that could be critical for understanding the broader physiological impacts of temperature changes on snapper. Additionally, the study focuses exclusively on thermal stress, which is indeed a significant factor in aquaculture, but does not consider other common stressors such as water quality, population density, or nutrition. In practical aquaculture settings, these factors often interact, influencing fish health in complex ways that may limit the direct applicability of the findings under multifactorial conditions. Moreover, the study observes fish responses over a three-month period, leaving open questions about the persistence of these effects. A longer-term follow-up could reveal whether chronic thermal stress has lasting impacts on gene expression or if the fish can fully recover once conditions return to normal. Extending the observation period might also provide insights into the potential for adaptation or resilience in response to prolonged stress. However, this study is methodologically robust and well-argued. While it focuses on a specific gene panel and thermal stress alone over a limited period, the Authors address these limitations effectively, providing a balanced and thoughtful discussion. Overall, this work lays a solid foundation for future studies to broaden genetic markers and environmental factors, making it a meaningful contribution to aquaculture research.

Response 1: Thank you for your positive comments on our study. We appreciate you highlighting that our work offers a meaningful contribution to aquaculture research.

Round 2

Reviewer 1 Report

Comments and Suggestions for Authors

The text overall is quite well prepared. My first comment would be to create some type of improved separation in Table 2 for each gene result. It is a table that is dense with text and could be more legible with horizontal lines or increased gaps between each gene result. 

Figure 2 still has "long term" and "short term" used inside the double-headed arrows for pdx5. The rbitrary placement of these double-headed arrows should be harmonized in some way to make the overall view to this figure more obvious. For example place them at the bottom of the X-axis for all graphs or at least the basal graphs for each group of graphs. 

It is a work of good interest for more susatainable aquaculture practices. 

Author Response

Comments 1: The text overall is quite well prepared. My first comment would be to create some type of improved separation in Table 2 for each gene result. It is a table that is dense with text and could be more legible with horizontal lines or increased gaps between each gene result. 

It is a work of good interest for more susatainable aquaculture practices. 

.

Response 1: Thank you for your positive comments. We have added horizontal lines between each gene result.

Comments 2: Figure 2 still has "long term" and "short term" used inside the double-headed arrows for pdx5. The rbitrary placement of these double-headed arrows should be harmonized in some way to make the overall view to this figure more obvious. For example place them at the bottom of the X-axis for all graphs or at least the basal graphs for each group of graphs. 

Response 2: Thank you for your comments. The edited version of the graph with chronic and acute in the double headed arrows was shown below in the last version. We have made further edits to the graph design as requested and placed the double headed arrows below the x axis for the graphs at the bottom.

Comments 3: It is a work of good interest for more susatainable aquaculture practices. 

Response 3: Thank you for your positive comment.

Reviewer 2 Report

Comments and Suggestions for Authors

I checked all the improvements apported to the manuscript, and in my honest opinion, I didn't find them satisfactory considering this paper worthy of the publications. 

The growth table provided (Table 1), even if it lacks a real temporal indication in the caption, it is reported that the data refers to the 3 different time points, however in the table the lines do not have this information. In any case the table reported that from the second to the third period both fish in the group “Cold treatment" only increase their biomass by 3 grams, and in the "Ambient" group they lost weight. Is this information correct? Both groups in the first period doubled their weight. Apart from this, the suggestion to add growth performance was to give further information to better discuss the results obtained, however, there is no mention of this data in the discussion, so they represent a useless addition.

Regarding the general observations, I still believe that this paper represents an appendix of the previous one, as the novelty of this research is inextricably linked to the data present in another paper (Bentley-Hewitt et al., 2024), without which, this manuscript has no strength to exist on its own. In accordance with this point, I checked the submission dates of the two papers, and it seems they were sent the same day. Hence due to the strict connections between the two works I wonder why the two results, given their similarity and the complete overlap of the genes analyzed, were not included in the same publication. Which would have made the analysis complete and non-repetitive.

For these reasons, I suggest reconsidering the manuscript and the data for a future submission.

Author Response

Comments 1:  I checked all the improvements apported to the manuscript, and in my honest opinion, I didn't find them satisfactory considering this paper worthy of the publications. The growth table provided (Table 1), even if it lacks a real temporal indication in the caption, it is reported that the data refers to the 3 different time points, however in the table the lines do not have this information. In any case the table reported that from the second to the third period both fish in the group “Cold treatment" only increase their biomass by 3 grams, and in the "Ambient" group they lost weight. Is this information correct?

Both groups in the first period doubled their weight. Apart from this, the suggestion to add growth performance was to give further information to better discuss the results obtained, however, there is no mention of this data in the discussion, so they represent a useless addition.

Response 1: Thank you for your comments. We have added an additional column to the table to indicate the sampling stage of the study. We can confirm that the increase in weight for the cold treatment over the 3 month study was 3.2 g and there was a very marginal decrease for the ambient treatment group (0.4 g). We have incorporated the growth data into the study results and discussion. We have added a new section into the results (3.1 Chronic-temperature experiment fish growth) lines 190-194. This section reads “

Snapper were sampled at 3 timepoints during the experiment. There were no large changes in fork length or weight between cold and ambient treatments groups. In comparison, fish became longer (P<0.001) and heavier (P<0.001) in the warm treatment group compared to the control at both mid-term and final sampling stages.“ and details of the statistical analysis is included in section 2.5 lines176-178, which reads “Fish phenotypic growth data were analysed using one-way analysis of variance (ANOVA) and Fisher’s least significant differences were calculated to compare treatment groups.”

In addition, we have included the following into the discussion on lines 292-330) “

First, we detected 24 significant changes in expression for snapper exposed to either warm (22°C) or cold (14°C) compared to ambient temperatures for 3 months. Because so many genes changed expression levels with the warm and cold treatments compared to the ambient temperature treatment across all tissue types, it was difficult to observe consistent trends in the dataset for some genes. This demonstrates the sensitivity of gene expression as a tool for monitoring the effects of chronic (long-term) changes in water temperature. The morphological data (Table 1) documenting changes in weight and fork length highlight that warmer temperatures are beneficial for snapper growth, in line with previous work on this species. Furthermore, we found that fish in the warm treatment group showed schooling behaviour throughout the experiment and were active during feeding events (fish from all treatment groups were fed ad libitum). In contrast, fish in the ambient and cold groups reduced their food intake and greatly reduced their movements, becoming almost stationary. These behavioural observations suggest a slowing of metabolism to conserve energy. Acceleration of weight gain with warmer temperature treatments has been previously observed in snapper in an experimental setting. [18]. It is difficult to speculate how much these differences in growth rate between treatment groups may have affected gene expression. However, it was notable that igf2 was significantly higher in the livers of snapper in the warm treatment group compared to the cold and ambient treatment groups. This gene is strongly correlated with the regulation of metabolism and is a growth factor in teleost fish. [32].”

Comments 2: Regarding the general observations, I still believe that this paper represents an appendix of the previous one, as the novelty of this research is inextricably linked to the data present in another paper (Bentley-Hewitt et al., 2024), without which, this manuscript has no strength to exist on its own. In accordance with this point, I checked the submission dates of the two papers, and it seems they were sent the same day. Hence due to the strict connections between the two works I wonder why the two results, given their similarity and the complete overlap of the genes analyzed, were not included in the same publication. Which would have made the analysis complete and non-repetitive.

For these reasons, I suggest reconsidering the manuscript and the data for a future submission.

Thank you for your comments. While we agree that the papers are clearly linked—hence our citation of the previous manuscript in the current submission—they differ significantly in their aims and scope, which are clearly separated.

  • Manuscript 1 introduces and describes the development of a novel stress and immune gene panel using NanoString technology for snapper. It further uses this panel to test and validate the technology by analysing gene expression in fin, head kidney, and liver tissues of snapper exposed to acute temperature changes (over 37 hours). This work identifies 10 key genes that could predict genotypes more tolerant to extreme temperatures.
  • Manuscript 2 (the current submission) assesses the impact of long-term (chronic) temperature changes over a 3-month period, using the gene panel developed in Manuscript 1. This represents a critical distinction, as acute temperature stress responses (Manuscript 1) differ significantly from long-term stress responses (Manuscript 2). Our findings strongly support this distinction.

Additionally, we conducted complementary analyses combining data from Manuscripts 1 and 2 to evaluate whether non-lethal fin clip samples could predict internal organ status (head kidney and liver). This exploratory work aims to assess stress resilience in wild stocks and aquaculture selective breeding programs, providing the study with greater statistical robustness.

We believe the current manuscript is not only substantially different from the previous one but also offers unique insights into long-term chronic temperature effects. Furthermore, it introduces the potential application of non-lethal fin clip sampling for monitoring fish health, a topic that was not covered in Manuscript 1.

Round 3

Reviewer 2 Report

Comments and Suggestions for Authors

I appreciated the effort carried out by the authors to fulfill the gaps present in the manuscript.